# Does Penicillin Allergy Increase the Risk of Surgical Site Infection after Orthognathic Surgery? A Multivariate Analysis

**DOI:** 10.3390/jcm11195556

**Published:** 2022-09-22

**Authors:** Eugénie Bertin, Christophe Meyer, Brice Chatelain, Aude Barrabé, Elise Weber, Aurélien Louvrier

**Affiliations:** 1Chirurgie Maxillo-Faciale, Stomatologie et Odontologie Hospitalière, CHU-Besançon, F-25000 Besançon, France; 2Department of Oral and Maxillofacial Surgery, University Hospital of Besançon, 3 Boulevard Fleming, CEDEX, F-25030 Besançon, France; 3Nanomedicine Lab, Imagery and Therapeutics, EA 4662, UFR Sciences et Techniques, Université de Bourgogne Franche-Comté, CEDEX, F-25030 Besançon, France

**Keywords:** orthognathic surgery, surgical site infection, penicillin allergy

## Abstract

This study aimed to demonstrate an association between the occurrence of surgical site infection (SSI) after orthognathic surgery and penicillin allergy and to assess whether other factors could be associated with the occurrence of SSI. A 10-year monocentric retrospective study was conducted to identify possible risk factors for SSI in orthognathic surgery. Bivariate analyses were performed using Fisher, Student, or Wilcoxon tests and multivariate analyses using logistic regression. Two hundred and sixty-six patients were included, and 3.5% had SSI. Bivariate analyses revealed a significant association between SSI and age at surgery (*p* = 0.01), penicillin allergy (*p* = 0.02), and postoperative antibiotic therapy by Clindamycin (Dalacine^®^) (*p* = 0.02). Multivariate analyses confirmed the association between the occurrence of SSI and treatment with Clindamycin (Dalacine^®^) or Clindamycin (Dalacine^®^) and Metronidazole (Flagyl^®^) postoperatively (*p* = 0.04). Antibiotic therapy with Clindamycin (Dalacine^®^) seems to be associated with a higher rate of SSI, and the mandible was the only site affected by SSI.

## 1. Introduction

Orthognathic surgery is performed to correct dentoskeletal disharmony. It has a functional and aesthetic aim but with a potential risk of complications. In the literature, the risk of surgical site infection (SSI) varies between 1.4 and 33.4% [1]. Many factors can contribute to the occurrence of SSI (diabetes, immunosuppression, smoking, age, type and duration of surgery, and type and duration of postoperative antibiotic therapy), and antibiotic prophylaxis is essential to act in the best interest of the patient without increasing the development of resistant strains.

There are no official recommendations for antibiotic prophylaxis in orthognathic surgery but the French Society of Anesthesia and Intensive Care Medicine (SFAR) recommends intraoperative antibiotic prophylaxis with Amoxicillin–Clavulanic Acid (Augmentin^®^), or Clindamycin (Dalacine^®^) in the case of allergy, for all surgeries with intraoral approaches [2]. Intraoperative antibiotic prophylaxis is commonly used, but whether or not to continue antibiotic therapy and, if so, for how long is a matter of debate. The protocols differ depending on the surgeon and department’s practices [3]. The literature shows a large number of practices: per-operative antibiotic prophylaxis only, short antibiotic therapy (less than 48 h), or prolonged antibiotic therapy (more than 3 days) [4,5]. Many studies have been published and all the results are different, making it difficult to reach a consensus. The antibiotic used also varies among studies. Amoxicillin–Clavulanic Acid (Augmentin^®^) is the most effective antibiotic treatment against oral micro-organisms, and few studies showed an increased risk of SSI in patients with penicillin allergy [6,7].

The objectives of this study were to demonstrate an association between the occurrence of SSI and penicillin allergy after orthognathic surgery and to assess whether other factors were associated with the occurrence of SSI.

## 2. Materials and Methods

A retrospective monocentric study (Department of Oral and Maxillofacial Surgery of the University Hospital of Besançon) was carried out. All patients who benefited from orthognathic surgery between 1 January 2012 and 1 April 2022 were identified using the French Social Security procedure list. The surgical procedures queried were Le Fort 1 osteotomy (LF1), bilateral sagittal split osteotomy (BSSO), genioplasty, a combination of BSSO + LF1, a combination of BSSO + LF1 + genioplasty, a combination of BSSO + genioplasty, a combination of LF1 + genioplasty, and surgically assisted rapid palatal expansion (SARPE).

We included all patients who had an osteotomy for dentoskeletal disharmony and whose records were complete. Patients who received an osteotomy not related to dentoskeletal disharmony or patients whose files were not complete were excluded.

For all included patients, we collected age at surgery, gender, smoking status, presence of diabetes or immune deficiency, presence of a penicillin allergy, type and duration of surgery, type and duration of postoperative antibiotic prophylaxis or therapy, and the occurrence of SSI. The diagnosis of SSI was defined by the presence of a purulent discharge or a purulent collection (clinical or radiological) in the surgical site, the occurrence of fever > 38.3°, or the secondary appearance of edema or local inflammatory signs. In the event of SSI, we noted the time of onset after surgery, the antibiotic prescribed and its duration, and the need for surgical drainage or not.

Statistical tests were carried out with R Studio software. Bivariate analyses to assess the relationship between the occurrence of SSI and the other variables were performed by a Fisher test for qualitative variables and a Student or Wilcoxon test (when the variable did not follow a normal distribution) for quantitative variables. For the multivariate analysis, we used logistic regression and included the variables for which there was a significant association in the bivariate analysis and variables for which there was an association in the literature or clinical relevance. All statistical tests considered a *p*-value of less than 0.05 to be significant.

The study was conducted in accordance with the Declaration of Helsinki and was approved by the ethics committee of the French Society of Stomatology, Maxillofacial Surgery and Oral Surgery (SFSCMFCO).

## 3. Results

Two hundred and ninety-five corresponding procedures were listed between January 2012 and April 2022. Thirteen were excluded because they were post-traumatic osteotomies, ten because they were the management of mandibular osteonecrosis, and six because the records were incomplete. A total of 266 patients were included in the study. Patients were systematically given intraoperative antibiotic prophylaxis with Amoxicillin–Clavulanic Acid (Augmentin^®^, 1 g if weight < 80 kg, 2 g if weight > 80 kg) or Clindamycin (Dalacine^®^ 600 mg) if the patient was allergic to penicillin. Four surgeons performed these procedures during the period studied. The surgeons used the same asepsis protocol with Betadine^®^, the same approach (intraoral), and the same osteosynthesis material: titanium plate secured with monocortical screws (Medartis, Basel, Switzerland). Patients undergoing BSSO and/or genioplasty received 3 days of 1 mg/kg corticosteroid (Methylprednisolone, Solumedrol^®^) therapy postoperatively.

The average age at the time of surgery was 26.5 years with a minimum of 14 years and a maximum of 57 years. The characteristics of the patients included and the type of surgeries performed are shown in Table 1. The immune deficiencies were hypersplenism (*n* = 1), trisomy 21 (*n* = 1), Crohn’s disease (*n* = 1), Turner’s syndrome (*n* = 1), macrophagic activation syndrome (*n* = 1), and familial Mediterranean fever (*n* = 1). The most frequently performed surgical procedures were BSSOs (37.2%), a combination of BSSO and LF1 (23.3%), and SARPE (18%). Concerning operating time, the average was 191 min with the shortest being 45 min and the longest being 475 min. The average duration of postoperative antibiotic therapy was 7 days (Figure 1). Regarding postoperative antibiotic therapy, 93.2% (*n* = 248) of the patients received Amoxicillin–Clavulanic Acid (Augmentin^®^) after surgery. Among the 18 patients with penicillin allergies, 66.7% (*n* = 12) received Clindamycin (Dalacine^®^) and 33.3% (*n* = 6) received Clindamycin (Dalacine^®^) and Metronidazole (Flagyl^®^). SSI was diagnosed in 3.5% (*n* = 9) of patients. The details of SSI are shown in Table 2. Among the patients who had an SSI, 66.7% (*n* = 6) underwent BSSO surgery and 33.3% (*n* = 3) a combination of BSSO and LF1 surgery. All SSI cases were in the mandible site (even in combination surgery). For the four patients who underwent surgical drainage, the bacteriological samples revealed germs of the commensal flora. No antibiogram was available because the bacteriology laboratory did not perform this test in the case of commensal flora without individualized germs. For the other patients who underwent surgical drainage, no samples were sent for analysis. For infected patients, early removal of the osteosynthesis material was not realized due to the favorable clinical evolution with antibiotic therapy and surgical drainage. No recurrence was observed after the first episode of SSI.

The occurrence of SSI was significantly associated with age at surgery (*p* = 0.01), with an average age of 26 years for patients who did not have SSI compared to 34 years for patients with SSI (Figure 2), with the presence of penicillin allergy (OR: 7.7; 95% CI: 1.1–40.8; *p* = 0.02) and with postoperative antibiotic therapy by Clindamycin (Dalacine^®^) (OR: 7.7; 95% CI: 1.1–40.8; *p* = 0.02). There was no significant association between the occurrence of SSI and gender (*p* = 0.3), smoking status (*p* = 1), diabetes (*p* = 1), immune deficiency (*p* = 1), type of surgery (*p* = 0.6), duration of surgery (*p* = 0.6), or duration of postoperative antibiotic therapy (*p* = 0.3).

For the multivariate analysis, we included age at surgery, the antibiotic prescribed postoperatively, smoking, diabetes, immune deficiencies, duration of postoperative antibiotic therapy, and type of surgery. We did not include penicillin allergy as this was the same data as the antibiotic prescribed postoperatively. The multivariate analysis showed a statistically significant association between the occurrence of SSI and treatment with Clindamycin (Dalacine^®^) or Clindamycin (Dalacine^®^) and Metronidazole (Flagyl^®^) postoperatively, with a relative risk multiplied by 2.3 (*p* = 0.04) compared to treatment with Amoxicillin–Clavulanic Acid (Augmentin^®^). The association with age at surgery was not confirmed by multivariate analysis (*p* = 0.1). There was also no association with diabetes (*p* = 0.9), smoking (*p* = 0.9), immune deficiencies (*p* = 0.9), the type of surgery (*p* = 0.3), or the duration of postoperative antibiotic therapy (*p* = 0.7) (Table 3).

## 4. Discussion

The study showed that penicillin allergy increases the relative risk of developing SSI by 2.3 times. Age initially appeared as a risk factor but was not confirmed by multivariate analysis, and the duration of postoperative antibiotic therapy was not found to be related to the occurrence of SSI.

There is no consensus on whether and for how long postoperative antibiotic prophylaxis should be continued. In reality, each team has its practices, and no study has shown an association between the occurrence of SSI and the duration of treatment. In our study, the duration varied between 2 and 15 days, with most patients receiving 5 to 10 days of antibiotics. Ghantous et al. [8], in their prospective double-blind, placebo-controlled study found no difference between 5 days of postoperative antibiotic therapy and 5 days of placebo. The results were the same for Baqain et al. [9] who also found no difference versus placebo over a 5-day period, but this study included only 17 patients per group. Others found no difference between intraoperative antibiotic prophylaxis alone and 3 days of postoperative antibiotic therapy [10] or between intraoperative antibiotic prophylaxis and 24 h of antibiotic therapy [11]. Davis et al. [12] found fewer SSIs with 48 h of postoperative antibiotic therapy vs. placebo. The duration of postoperative antibiotic therapy appears to be relatively long in our study. These durations are based on the habits of each of the four surgeons. However, the majority of patients received 5 days of antibiotic therapy, and this is consistent with the data in the literature and the habits of other centers. The practices and data in the literature are diverse and the results varied, but it would seem that a maximum of 5 days of antibiotic therapy is sufficient to prevent the occurrence of SSI.

In our study, penicillin allergy appears to be a risk factor for SSI, and these data seem to be in line with the literature: relative risk multiplied by 2.6 in patients treated with Clindamycin (Dalacine^®^) in the study of Roistacher et al. [6], and there is a decrease in SSI in patients treated with Amoxicillin–Clavulanic Acid (Augmentin^®^) in the study of Barrier et al. [7]. To remedy this, it appears necessary, first, to identify true allergies to penicillin. Many patients who have been labeled as allergic are not so [13]. For “true” allergies, all studies use Clindamycin (Dalacine^®^), as recommended, but other antibiotics may be more effective in preventing SSI in orthognathic surgery. Recommendations are in favor of using Clindamycin (Dalacine^®^) in the case of allergy. The systematic addition of Metronidazole (Flagyl^®^) could allow better antibiotic coverage with protection against anaerobic germs.

No statistical relationship was found between the occurrence of SSI and the type of surgery, but in this study, all infected patients had undergone BSSO or a combination of BSSO and LF1 surgery, and the site of infection was always the mandible. Van Camp et al. [3] found the same results with SSI only in patients who received BSSO or a combination of BSSO and LF1 surgery with no statistically significant association. SSI most frequently occurred at the BSSO incision for Davis et al. [12]. In another study by Davis et al. [14], more SSIs were found if the operation involved two sites and if it involved the mandible. In the study by Cousin et al., the most common site of infection was the mandible but no statistical comparison was made [15]. The mandible appears to be more prone to infection than the maxilla even if there is no statistical relationship.

Regarding other potential risk factors, patients with SSI appeared to be older in our study, although this relationship was not confirmed by multivariate analysis. These results are similar to the study by Bouchard et al. [16]. However, in both studies, the age difference was small and involved relatively young patients. Diabetes, immune deficiencies, and smoking are recognized risk factors for SSI in surgery. In this study, patients with diabetes or immune deficiencies represent too small a sample to obtain meaningful results. For patients who smoke, the rate is higher, but the association is probably not significant because patients are forced to stop smoking in the weeks preceding and following the procedure.

The main limitation of our study is that the number of SSIs is small, and therefore the results cannot be extrapolated. The definition of an SSI was not standardized as it might have been in the case of a prospective study. In our study, we defined SSI as an infection related to a surgical procedure that occurs near the surgical site within one year following surgery because an implant is involved [17]. The SSIs occur relatively late after surgery (at 15 days postoperatively and the others from the first month): this may be explained by lesser compliance with dietary and hygienic instructions when moving away from the surgery and with the disappearance of pain. A larger sample of patients is needed to make real inferences. In addition, patients’ compliance with dietary and oral hygiene instructions could not be assessed. Similarly, this is a retrospective study with data collection by file only. There is a potential bias in data collection that depends on the accuracy noted in the medical records.

It appears that the presence of penicillin allergy is a risk factor for SSI, and all cases of SSI were found in the mandible despite the lack of significance of the results. Regarding the duration of postoperative antibiotic therapy, a duration longer than 5 days does not seem justified. A prospective study comparing different durations of antibiotic therapy with a large number of patients could provide more precise results and change practices.

## Figures and Tables

**Figure 1 jcm-11-05556-f001:**
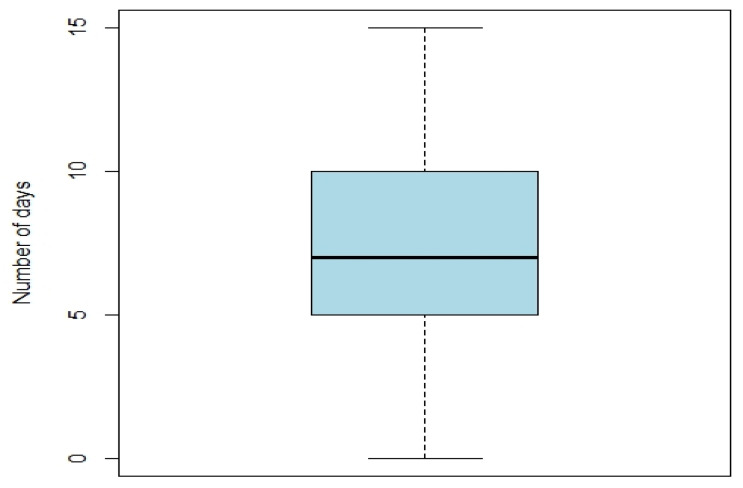
Duration of postoperative antibiotic therapy.

**Figure 2 jcm-11-05556-f002:**
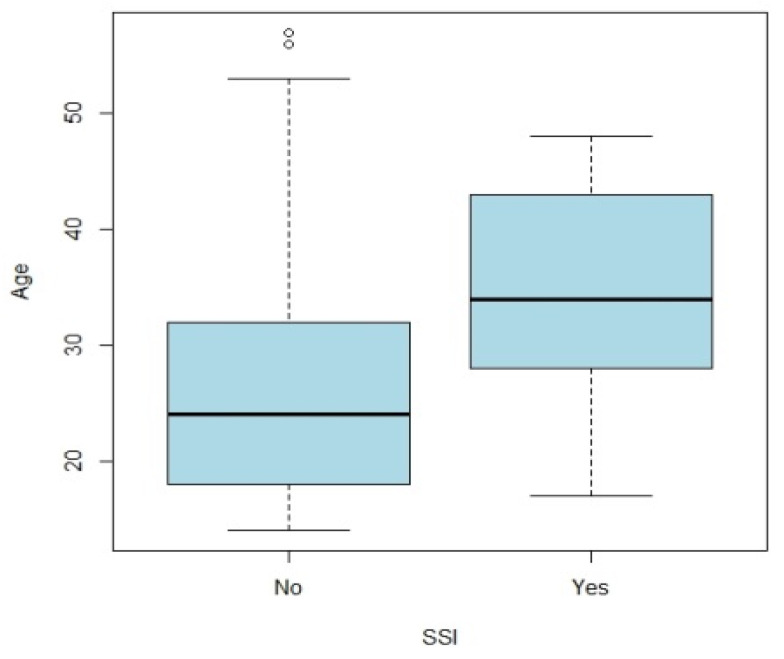
Association between age at surgery and SSI.

**Table 1 jcm-11-05556-t001:** Characteristics of the patients and type of surgeries.

	Number of Patients	%
**Gender**		
- **Male**	91	34.2
- **Female**	175	65.8
**Active smoking**		
- **No**	211	79.6
- **Yes**	54	20.4
**Diabetes**	2	0.7
**Immunodeficiency**	6	2.2
**Penicillin allergy**	18	6.8
**Type of surgeries**		
- **BSSO**	99	37.2
- **BSSO + LF1**	62	23.3
- **SARPE**	48	18
- **BSSO + LF1 + genioplasty**	20	7.5
- **BSSO and genioplasty**	16	6
- **Genioplasty**	13	4.9
- **LF1**	6	2.3
- **LF1 and genioplasty**	2	0.8

BSSO: bilateral sagittal split osteotomy, SARPE: surgically assisted rapid palatal expansion, LF1: Le Fort 1 osteotomy.

**Table 2 jcm-11-05556-t002:** Details of Surgical Site Infection.

Time after Surgery (Days)	Type of Drainage	Antibiotic
60	General anesthesia	Amoxicillin–Clavulanic Acid (Augmentin^®^) and Gentamycin
150	Local anesthesia	Clindamycin (Dalacine^®^)
30	Local anesthesia	Amoxicillin–Clavulanic Acid (Augmentin^®^)
180	General anesthesia	Clindamycin (Dalacine^®^)
15	Local anesthesia	Amoxicillin–Clavulanic Acid (Augmentin^®^)
90	Local anesthesia	Amoxicillin–Clavulanic Acid (Augmentin^®^)
30	Local anesthesia	Amoxicillin–Clavulanic Acid (Augmentin^®^)
45	None	Amoxicillin–Clavulanic Acid (Augmentin^®^)
15	None	Clindamycin (Dalacine^®^)

**Table 3 jcm-11-05556-t003:** Multivariate analysis.

	RR	*p*
Age	6	0.1
Active smoking	−1.8	0.9
Diabetes	−1.3	0.9
Immune deficiencies	−1.7	0.9
Type of surgery	8.2	0.3
Postoperative antibiotic therapy	**2.3**	**0.04**
Duration of postoperative antibiotic therapy	0.05	0.7

RR: relative risk. *p* < 0.05 was considered statistically significant.

## Data Availability

Not applicable.

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
