# Peer review of "Does Penicillin Allergy Increase the Risk of Surgical Site Infection after Orthognathic Surgery? A Multivariate Analysis"

_jcm, 2022, doi:10.3390/jcm11195556_

Round 1

Reviewer 1 Report

Thank you for the opportunity to review this article.

As a whole, it is an important issue and was well checked and presented.

I would like to highlight a few issues for minor revision before acceptance:

Introduction:

line 31: either provide all the factors or add "the most significant".

Materials and methods:

line 52: I think there is and improper translation from French here: "Maxillofacial" not "Maxillo-facial".

line 55-56: Bimaxillary osteototomy is an outdated term and it is not a procedure for itself but a combination of LF1 osterotomy with BSSO or other mandibular osteotomy.

line 63: not all antibiotic consumption is "prophylaxis"

Results:

line 95 - which corticosteroid?

line 109 - re-phrase "managed"

line 110 - BSSO surgery not "surgery for BSSO" as it's not a condition

line 111 - which mandibular osteotomy in two-jaw surgery?

Table 1 - Specify which two-jaw surgery

Figure 1: Legend is in French

Table 2 - The infection measured in days or hours? Does 180 days postoperative is considered SSI? I think it's too long.

lines 130-136 - what are the values that were compared and gave the statistical significance?

lines 141-152 - table that shows comparison values in needed

Discussion: 

Overall improvement of English is needed. Where is conclusion and what is it? Limitations section need to be greatly expanded.

Author Response

Dear Editors and Reviewers,

Thank you for reviewing and commenting on our article entitled: “Does penicillin allergy increase the risk of surgical site infection after orthognathic surgery? A multivariate analysis” we would like you to reconsider for publication in your Journal. Please find here our revised article and below the responses to the reviewers’ comments.

Introduction:

The list of factors has been completed (line 31)

Materials and methods:

The text has been corrected:

- “Maxillo-facial” by “Maxillofacial” (line 52)

- “bimaxillary osteotomy” by “BSSO + LF1” (line 55-56)

The list of procedures were rewritten: “Le Fort 1 osteotomy (LF1), bilateral sagittal split osteotomy (BSSO), genioplasty, a combination of BSSO + LF1, a combination of BSSO + LF1 + genioplasty, a combination of BSSO + genioplasty, a combination of LF1 + genioplasty and Surgical Assisted Rapid Palatal Expansion (SARPE).”

The term antibiotic “prophylaxis” has been completed by “prophylaxis or therapy” (line 63)

Results:

We added the type of corticosteroid: “Methylprednisolone, Solumedrol®” (line 95)

“managed” was replaced by “diagnosed” (line 109)

“surgery for BSSO” was replaced by “BSSO surgery” (line 110)

We have developed the type of BSSO in two-jaw surgery (line 111)

Figure 1: legend was corrected: “Duration of postoperative antibiotic therapy.”

Regarding the time to occurrence of surgical site infection, we considered it to be SSI up to one year postoperatively because an implant was placed in accordance with the article of Di Benedetto et al. (C. Di Benedetto, A. Bruno, et E. Bernasconi, « Surgical site infection: risk factors, prevention, Diagnosis and treatment », Rev Med Suisse, vol. 9, no 401, p. 1832‑1834, 1836‑1839, Oct. 2013). We consider that, if a surgical procedure with an implant had not been performed, the patient would most likely not have had an infection. We understand this is considered a late infection. We did not find any infection after one year because in our practice we systematically remove the material one year after the initial surgery. We have therefore deliberately wished to show this result, even if it is open to criticism.

We have added the precise results of the statistical tests (line 130-136)

A table representing the results of the multivariate analysis was added: “table 3”

Discussion:

We have proofread and edited the English (with an English native speaker), expanded the section on limitations and added a conclusion.

All authors have approved the corrected manuscript.

Please receive our bests regards.

Dr. Eugénie BERTIN

Reviewer 2 Report

Comment

Thanks the authors to submit a comprehensive study addressing “Does penicillin allergy increase the risk of surgical site infection after orthognathic surgery? A multivariate analysis.” This is an interesting topic to discuss multiple factors related to surgical site infection following orthognathic surgery.

1.      Could you please list a table to show the findings of multivariate analysis, which will clearly and easily to elaborate all the relevant information to the readers.

2.      Figure 1, please switch the French to English.

3.      In table 1, please list the abbreviations to the below of the table.

4.      As a reader, I would like to know the types of bacteria cultured in this series to cause the surgical site infection. Could you provide the data, which will benefit our readers to prevent the SSI in their practice.

5.      A concise conclusion paragraph is necessary. Please add this content.

6.      How to deal with the cases with SSI in your experiences, the readers are interested in the way you provided. To remove the plates and screws or not, and when is the optimal time to do this, please discuss.

Pease take care of the above suggestions, major revision is needed.

Author Response

Dear Editors and Reviewers,

Thank you for reviewing and commenting on our article entitled: “Does penicillin allergy increase the risk of surgical site infection after orthognathic surgery? A multivariate analysis” we would like you to reconsider for publication in your Journal. Please find here our revised article and below the responses to the reviewers’ comments.

For clearly and easy comprehension, we have added a table with the specific results of the multivariate analysis (“Table 3”).

Figure 1: legend was corrected: “Duration of postoperative antibiotic therapy.”

Abbreviations were listed below in the table “BSSO: Bilateral Sagittal Split Osteotomy, SARPE: Surgical Assisted Rapid Palatal Expansion, LF1: Le Fort 1 osteotomy”.

As regards the germs involved in SSIs, this very interesting data have been completed. These were commensal flora for the cases for which bacteriological analyses were carried out. No specific germ was identified.

A concluding paragraph has been added.

We have developed a part on the management of our SSI. No patient benefited from the early removal of the osteosynthesis material. Indeed, the initial management by antibiotic therapy alone, associated or not with surgical drainage, allowed a favourable evolution for all patients.

We have proofread and edited the English (with an English native speaker).

All authors have approved the corrected manuscript.

Please receive our bests regards.

Dr. Eugénie BERTIN

Round 2

Reviewer 2 Report

Thank you for revision.

Author Response

Dear reviewer,

Thank you for your comments and suggestions on our work.

Bests regards

Dr Eugénie Bertin